# AUGMENTATION CURRICULUM LEARNING FOR GENERALIZATION IN REINFORCEMENT LEARNING

## ABSTRACT

Many Reinforcement Learning tasks rely solely on pixel-based observations of the environment. During deployment, these observations can fall victim to visual perturbations and distortions, causing the agent's policy to significantly degrade in performance. This motivates the need for robust agents that can generalize in the face of visual distribution shift. One common technique for doing this is to apply augmentations during training; however, it comes at the cost of performance. We propose Augmentation Curriculum Learning a novel curriculum learning approach that schedules augmentation into training into a weak augmentation phase and strong augmentation phase. We also introduce a novel visual augmentation strategy that proves to aid in the benchmarks we evaluate on. Our method achieves state-of-the-art performance on Deep Mind Control Generalization Benchmark.

## 1 INTRODUCTION

Reinforcement Learning (RL) has shown great success in a large variety of problems from video-games Mnih et al. (2013), navigation Wijmans et al. (2019), and manipulation Levine et al. (2016); Kalashnikov et al. (2018) even while operating from high-dimensional pixel inputs. Despite this success, the policies produced by RL are only well suited for the same environment they were trained for and fail to generalize to new environments. Instead, agents overfit to task-irrelevant visual features, resulting in even simple visual distortions degrading policy performance. A key objective of image-based RL is building robust agents that can generalize beyond the training environment.

Several existing approaches to training more robust agents include domain randomization Pinto et al. (2017); Tobin et al. (2017) and data augmentation Hansen & Wang (2021); Hansen et al. (2021b); Fan et al. (2021). Domain randomization modifies the training environment simulator to create more varied training data, whereas data augmentation deals with augmenting the image observations representing states without modifying the simulator itself. Prior work shows pixel-based augmentation improves sample efficiency and helps agents achieve performance matching state-based RL Laskin et al.; Yarats et al. (2021). Therefore, in this work, we focus on data augmented generalization to visual distribution shift while the semantics remain unchanged. We specifically aim to do this in a zero-shot manner, i.e., where shifted data is unavailable during training.

Unlike for supervised and self-supervised image classification tasks, augmentation for pixel-based RL has demonstrated mixed levels of success. Prior work categorized augmentations into weak and strong ones based on downstream training performance Fan et al. (2021). Specifically, works define **weak augmentations** as those allowing the agent to learn a policy with higher episodic rewards in the training environment than training without augmentation. **Strong augmentations** refers to augmentations that lead to empirically worse performance than training with no augmentations. Classifying augmentations according to this definition is dependent on the task. For example, cut-out color has empirically been shown to be detrimental ("strong augmentation") for all tasks in Deep Mind Control Suite (DMC) Tassa et al. (2018), but is a effective ("weak augmentation") for Star Pilot in Procgen Cobbe et al. (2019) as shown in Laskin et al.. Methods exist that attempt to automate finding the optimal weak augmentation on a per-task basis Raileanu et al. (2020), but these still do not expand the effectiveness of many augmentations.

Many RL generalization methods leverage weak augmentation for better policy learning training and add strong augmentations in training for generalization to visual distribution shift Hansen & Wang (2021); Hansen et al. (2021b); Fan et al. (2021). However, these methods suffer from strong

augmentation making training harder due to the difficulty of learning from such diverse visual observations, destabilizing training. This results in strong augmentations causing the agent to not learn a policy with as strong performance as using weak augmentations alone. In this work, we introduce a new training method that avoids the training instabilities caused by strong augmentations through a curriculum that separates augmented training into weak and strong training phases. Once the network has been sufficiently regularized in the weak augmentation phase, it is cloned to create a policy network that is trained on strong augmentations. This disentangles the responsibilities of the networks into accurately approximation the Q-value (network trained on weak augmentations) of the agent and generalization (doing well on shifted test distributions). Crucially we separate the two networks to avoid the destabilizing effect of strong augmentations. We also demonstrate the power of the method under even more severe augmentation, namely a new splicing augmentation that pastes relevant visual features into an irrelevant background. We show that our curriculum learning approach can effectively leverage strong augmentations, and the combination of our method with this new augmentation technique achieves state-of-the-art generalization performance.

**Our main contributions are summarized as follows:**

• We introduce Augmentation Curriculum Learning (AugCL), a new method for learning with strong visual augmentations for generalization to unseen environments in pixel-based RL.

• A new visual augmentation named Splice, which by simulating distracting backgrounds helps prevent overfitting to task irrelevant features.

• We demonstrate AugCL achieves state-of-the-art results across a suite of pixel-based RL generalization benchmarks.

## 2 RELATED WORKS

### 2.1 CURRICULUM LEARNING

Inspired by how humans learn, (Elman, 1993) proposed training networks in a curriculum style by starting with easier training examples and then gradually increasing complexity as training ensues. (Bengio et al., 2009) showed that this training style yielded better generalization results faster, and by introducing more difficult examples gradually, online training could be sped up. This ideology has shown to be transferable to RL across varying types of generalization (Cobbe et al., 2019), (Wang et al., 2019), (Florensa et al., 2018), (Sukhbaatar et al., 2017), but to our knowledge has never been explored for generalization to visual perturbations in pixel-based RL.

### 2.2 RL GENERALIZATION BENCHMARKS

There are many benchmarks designed for evaluating agents under different distribution shifts (Chattopadhyay et al., 2021; Dosovitskiy et al., 2017; Stone et al., 2021; Zhu et al., 2020; Li et al., 2021; Szot et al., 2021). We chose Deep Mind Control Generalization Benchmark (DMC-GB) (Hansen & Wang, 2021) as the current SOTA methods have been benchmarked on this, allowing us to compare directly to the results shown in previous works. DMC-GB offers 4 different generalization modes: color easy, color hard, video easy, and video hard. These modes can be applied to all DMC tasks, and visual examples can be seen in Figure 4. Color easy is not benchmarked on as it is considered solved (Hansen & Wang, 2021). Color hard dynamically changes the color of the agent, background, and flooring. Video easy changes the background to another random image, and video hard will change both the background and floor to a random image. Under these extreme perturbations, the agent must learn to identify the relevant visual features to the task in order to maximize reward.

### 2.3 GENERALIZATION IN VISUAL RL

There have been many advances in visual RL generalization. In this section, we will briefly summarize each method. SODA (Hansen & Wang, 2021) leverages an approximate contrastive loss focused on minimizing the distance of embedded vectors of the same state augmented with crop and a strongly augmented copy closer together in hyper-dimensional space. SECANT (Fan et al., 2021) trains an agent using crop and then leverages that agent's policy for training a new agent under strong augmentation in an imitation learning fashion. The prior SOTA approach to DMC-GB was

SVEA (Hansen et al., 2021b). SVEA leverages data-mixing, which modifies the critic loss Equation (1) to a weighted weak augmented and strong augmented temporal difference error (Sutton & Barto, 2018). We take inspiration from the SVEA in our method by bootstrapping a Q target network that has gradient updates under weak augmentation for a critic trained under strong augmentation.

# 3 BACKGROUND

## 3.1 PROBLEM: GENERALIZATION IN PIXEL CONTROL TASKS

We formulate our problem as interacting with a Markov Decision Process (MDP) defined as the tuple $\mathcal{M} = (\mathcal{S}, \mathcal{A}, \mathcal{P}, \mathcal{R}, \gamma)$ for state-space $\mathcal{S}$, action space $\mathcal{A}$, transition distribution $\mathcal{P}(s'|s, a)$, reward function $\mathcal{R}(s, a)$, and discount factor $\gamma$. In our setting, images $o \in \mathcal{O}$ offer only partial observability. We therefore represent the state with $k$ stacked observations to maintain the Markov property. The goal is to learn a policy $\pi(a_t|s_t)$ which maps states to a distribution over actions to maximize $R = \mathbb{E}_{\tau \sim \pi} \left[ \sum_{t=1}^{T} \gamma^t \mathcal{R}(s_t, a_t) \right]$.

In this work we focus on learning a policy that generalizes to new MDPs $\overline{M}$. $\overline{M}$ has a new observation space, but other elements of the MDP are unchanged. Policy $\pi$ is then evaluted on the return on new MDPs, without any additional samples from the new MDPs for updating $\pi$.

## 3.2 SOFT ACTOR-CRITIC

SAC (Haarnoja et al., 2017) is considered the state-of-the-art (SOTA) off-policy RL algorithm for most continuous control tasks. SAC trains an actor network $\pi_\psi(a_t|s_t)$ to take actions and critic network $Q_\phi(s_t, a_t)$ to predict state-action values. SAC follows the maximum entropy RL principle which is to maximize actor entropy while maximizing expected reward $\mathbb{E}_{s_t, a_t \sim \pi}[\sum_t r_t + \alpha \mathcal{H}(\pi(\cdot|s_t))]$. The critic is trained by minimizing the temporal difference error using samples $\tau_t = (s_t, a_t, s_{t+1}, r_t) \sim D$, where $D$ is a replay buffer storing the agent's previous interactions with the environment during training. The critic parameters can be updated using an approximation to the expected rewards. Let $\phi$ be the critic parameters and $\phi^{target}$ be another set of critic parameters for the bootstrapped value target.

$$\mathcal{L}_Q(\phi) = \mathbb{E}_{\tau \sim D}[(Q_\phi(s_t, a_t) - (r_t + \gamma V(s_{t+1}; \phi^{target})))^2] \tag{1}$$

The value of the next state is estimated through the single-step boostrap target based on the target critic network parameters:

$$V(s_{t+1}; \phi^{target}) = \mathbb{E}_{a' \sim \pi}[Q_{\phi^{target}}(s_{t+1}, a') - \alpha \log \pi_\psi(a'|s_{t+1})] \tag{2}$$

Target critic parameters $\phi^{target}$ are typically updated as an exponential moving average (EMA) of the regular critic parameters $\phi$. The actor network learns a policy by minimizing the following.

$$\mathcal{L}_\pi(\psi; \phi) = -\mathbb{E}_{a \sim \pi}[Q_\phi(s_t, a) - \alpha \log \pi_\psi(a|s_t)] \tag{3}$$

Typically, $\pi_\psi$ is represented by a Gaussian distribution for continuous control and updates are made via the reparameterization trick (Kingma & Welling, 2013) and $\alpha$ is learned in relation to entropy. All methods in the experiments section 5 use the same base neural architecture and SAC with EMA updates of Q-target network from the critic network. A key thing to note is that the actor and critic share the same CNN encoder. This is crucial to the success of our's and previous methods. More details in appendix C.

# 4 METHOD

## 4.1 AGENT ARCHITECTURE

The key differences between AugCL and previous works are: **1)** We train a weak and strong augmented network in parallel. **2)** We train only on weak augmentation for the early phases of training. **3)** We bootstrap the strong augmented network from a weakly augmented Q target network. The driving intuitions behind this is strong augmentations incur non-zero degradation to policy learning but help with generalization. Hence to mitigate this issue, we have a separate network trained only on

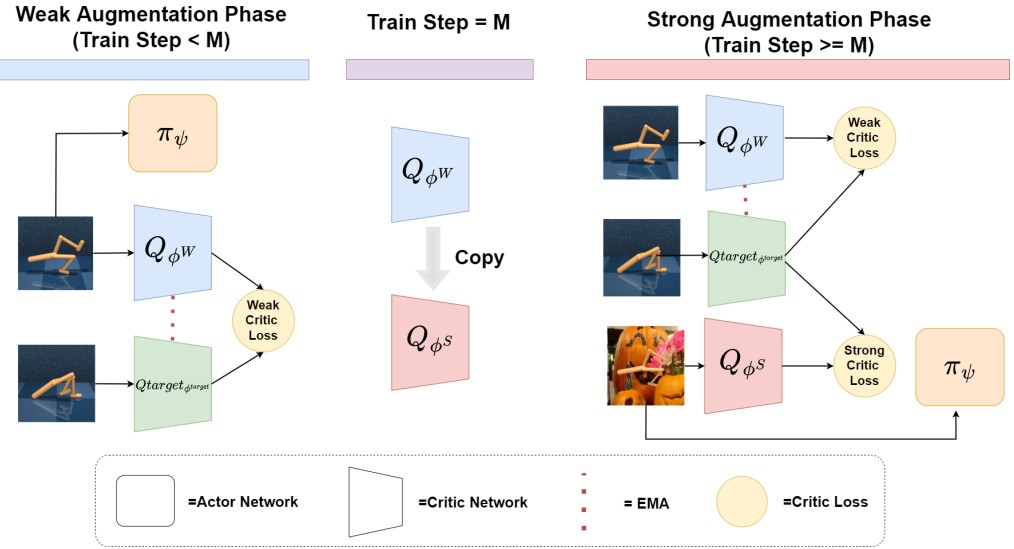

Figure 1: Neural architecture and tensor flow across different phases for AugCL.

strong augmentation and update the target network using EMA from the weak augmented network. This duo circumvents the policy degradation by allowing the weak critic and Q target network to learn a state and action value approximation without being impeded by strong augmentation. Then by bootstrapping this Q target network with an accurate approximation of the value function, the strong augmented network learns to generalize under strong augmentation. The weak augmented pre-training is required to regularize the CNN from biasing itself to high-frequency features. We show both are necessary to achieve SOTA performance in Section 5.3.

AugCL builds off of SAC, learning a critic and policy network as shown in Figure 1. The critic network $Q_\phi(s_t, a_t)$ takes as input a stacked sequence of image observations, and an action produces the expected return for taking action $a_t$ in state $s_t$. We also learn a parameterized policy $\pi_\psi(s_t)$ that outputs a normal distribution parameterized by a learned mean and variance which then samples an action for the current state. $Q_\phi$ and $\pi_\psi$ share a visual encoder $h_\mu(s_t)$ which takes as input the high-dimensional image observations and produces a low-dimensional state encoding. The parameters of $\psi$ and $\phi$ both include the encoder parameters $\mu$, which are updated as a part of both the policy and critic losses.

AugCL trains $Q_\phi$ and $\pi_\psi$ through the standard SAC losses with random image augmentations. We sample augmentations $f$ from a distribution over augmentations $\mathcal{F}$. We then use $f$ to augment the states $s_t$ in the SAC losses from Section 3.2. We augment only the current state $s_t$ in the SAC loss similar to (Hansen et al., 2021b) since we found this improves stability as shown in sec B. The resulting losses for the critic and policy with the augmentations are respectively:

$$\mathcal{L}_Q(\phi; \phi^{target}, \mathcal{F}) = \mathbb{E}_{\tau \sim D, f \sim \mathcal{F}}[(Q_\phi(f(s_t), a_t) - (r_t + \gamma V(s_{t+1}; \phi^{target})))^2] \quad (4)$$

$$\mathcal{L}_\pi(\psi; \phi, \mathcal{F}) = -\mathbb{E}_{a \sim \pi, f \sim \mathcal{F}}[Q_\phi(f(s_t), a) - \alpha \log \pi_\psi(a|f(s_t))] \quad (5)$$

Notice that the losses are defined relative to a distribution over augmentations $\mathcal{F}$. The choice of this augmentation distribution is an important consideration in the algorithm's stability and robustness to new MDPs. Prior work breaks up augmentations for pixel-based control into two classes: weak augmentations and strong augmentations.

**Weak Augmentations** (denoted as $\mathcal{F}^W$) are augmentations that help stabilize and improve training performance in the source MDP $\mathcal{M}$, but are insufficient for generalization to new MDPs $\overline{\mathcal{M}}$. (Cetin et al., 2022) shows that training with weak augmentations is important to prevent overfitting to high-frequency features in the image space when learning from bootstrapped targets in actor-critic methods. The consequences of this overfitting have been shown to cause the critic network to overfit to its own predictions, and a decrease in correlation to Monte Carlo returns as training ensues. Training with weak augmentations is therefore an important part of any actor-critic control-from-pixels

method, such as AugCL. Known weak augmentations for all DMC tasks are: crop, translate (Laskin et al.) and shift (Yarats et al., 2021).

**Strong Augmentations** (denoted as $\mathcal{F}^S$) are augmentations that help improve policy performance in new MDPs $\overline{\mathcal{M}}$. Visual perturbations such as random color changes of the environment can be simulated with strong augmentations such as: random convolution and random color jitter. While distracting backgrounds can be simulated with mix-up (Zhang et al., 2017), these augmentations are difficult to train with, as they increase the difficulty of the learning problem due to the duo of stochastic parameter sampling for $\mathcal{F}^S$ and high visual variance between samples. AugCL addresses how to effectively incorporate strong augmentations such as: random convolution, overlay (variation of mix-up), and our novel augmentation into training.

## 4.2 AugCL: Curriculum Learning with Strong Augmentations

As mentioned, strong augmentations are detrimental to learning but important to train with for generalization performance. The key idea of our method is therefore to leverage curriculum learning (Bengio et al., 2009) to avoid this destabilization. Specifically, AugCL defines a curriculum over augmentations to enable better training and generalization. It is well known that CNNs are inherently biased to high-frequency features (Geirhos et al., 2018; Jo & Bengio, 2017), the consequences of this in RL is a lower average episodic reward in the train environment. AugCL uses weak augmentations early in training to regularize the CNN. Then later in training AugCL introduces strong augmentations to improve the robustness of the policy. We train two separate networks in parallel as we believe that strong augmentation incurs a non-zero degradation to policy learning, as shown in Section 5.3.

---

**Algorithm 1** AugCL

---

1:  $\phi^W, \psi, \phi^{target}$ : critic parameters, policy parameters and Q-target parameters
2:  $\alpha, \beta, \zeta$: actor learning rate, critic learning weight and momentum encoder weight
3:  $M$: Update step to switch to strong augmentation
4:  **for** timestep $t = 1, ..., T$ **do**
5:  $\quad a_t \sim \pi_\psi(\cdot|s_t)$ $\hfill \triangleright$ Sample action
6:  $\quad s_{t+1} \sim \mathcal{P}(\cdot|s_t, a_t)$ $\hfill \triangleright$ Step environment
7:  $\quad \mathcal{B} \leftarrow \mathcal{B} \bigcup (s_t, a_t, r(s_t, a_t), s_{t+1})$ $\hfill \triangleright$ Add transition to replay buffer
8:  $\quad$ **if** $t = M$ **then**
9:  $\quad\quad \phi^S = \phi^W$ $\hfill \triangleright$ Clone critic
10: $\quad \{s_i, a_i, r_i, s_{i+1}|i = 1, ..., N\} \sim \mathcal{B}$ $\hfill \triangleright$ Sample transition
11: $\quad \phi^W \leftarrow \phi^W - \beta \nabla_\phi \mathcal{L}_Q(\phi^W; \mathcal{F}^W, \phi^{target})$ $\hfill \triangleright$ Weak Critic loss
12: $\quad$ **if** $t \geq M$ **then**
13: $\quad\quad \psi \leftarrow \psi - \alpha \nabla_\psi \mathcal{L}_\pi(\psi; \mathcal{F}^S)$ $\hfill \triangleright$ Strong Actor loss
14: $\quad\quad \phi^S \leftarrow \phi^S - \beta \nabla_\phi \mathcal{L}_Q(\phi^S; \mathcal{F}^S, \phi^{target})$ $\hfill \triangleright$ Strong Critic loss
15: $\quad$ **else**
16: $\quad\quad \psi \leftarrow \psi - \alpha \Delta_\psi \mathcal{L}_\pi(\psi; \mathcal{F}^W)$ $\hfill \triangleright$ Weak Actor loss
17: $\quad \phi^{target} \leftarrow (1 - \zeta)\phi^{target} + \zeta\phi^W$ $\hfill \triangleright$ Q-target EMA update

---

AugCL is described in Algorithm 1. AugCL begins by acting in the environment with $\pi_\psi$ and then adding the observed transition to the replay buffer (lines 5-7). We then sample data batches from the replay buffer for updating the policy and critic. Our curriculum learning schedule breaks the updates into two phases. For the first $M$ policy updates, AugCL is in the *weak augmentation phase* and updates the critic and policy from weak augmentations alone (lines 11,16). Target critic parameters $\phi^{target}$ are updated as exponential moving averages of the learned critic parameters $\phi$ (line 17), and used in the bootstrap term of the critic loss. The purpose of the weak augmentation phase is to stabilize policy learning. Prior work shows that it is easy for the critic to overfit in image-based RL and weak augmentations are important to achieve strong training performance (Cetin et al., 2022). However, the weak augmentations do not make the policy robust to new visuals. Improving generalization performance is the purpose of the next *strong augmentation phase*.

Then, after $M$ policy updates, AugCL switches to the *strong augmentation phase* and incorporates strong augmentations into training (lines 12-14). A new strong critic network $\phi^S$ is copied from

the weak critic network $\phi^W$ (line 9). Now, the weak critic network is updated like in the previous phase by training with weak augmentations. However, the separate strong augmentation network with parameters $\phi^S$ is now trained with strong augmentations (line 14). The policy is also updated with the strong augmentations (line 1) . Separating the strong and weak augmentations into two networks is important for stability. The weak critic helps stabilize bootstrap targets by leveraging weak augmentation to better approximate the state, action value function while the strong critic focuses on generalization performance. Previous methods have attempted this parallel training of the weak and strong augmented network, but with little success (Fan et al., 2021). We show that this is due to not regularizing the CNN encoder first in sec 5.3. A figure of the architecture and the flow of tensors representing input and output of each neural layer can be seen in Figure 1.

The advantage of AugCL separating strong augmentations into a later phase of learning is it does not require a delicate balance between potentially conflicting losses from strong and weak augmentations. SODA and SVEA incorporate strong augmentations as an auxiliary learning signal that is always applied in conjunction with learning an accurate approximation of the state, action value from weakly augmented data. By learning from both data at the same time, the networks must contend with the trade off between stronger augmentations improving generalization yet harming training performance. The auxiliary objective in SODA may suffer from gradient interference from the conflicting losses as the critic network is optimized to learn an accurate state, action value approximation, and a contrastive loss in parallel. SVEA suffers from a similar issue in that it requires a hyperparameter to balance the combination of losses from strong and weak augmented data. On the other hand, AugCL disentangles the responsibilities of state, action value approximation, and generalization between the weak augmented critic and the strong augmented critic respectively. We empirically demonstrate this by showing AugCL performs better than SVEA and SODA on a variety of benchmarks and has minimal train environment performance degradation, as shown in Appendix D. Also note SODA and SVEA only pass weakly augmented observations to the policy network during training, the issue with this is that $\pi_\phi(\mathcal{F}^W(s_t)) \neq \pi_\phi(\mathcal{F}^S(s_t))$. AugCL does not suffer from this issue as we pass $\mathcal{F}^S(s_t)$ through the policy network at train time, and interestingly we find train performance still improves and converges to a marginally worse performance on the train environment than training with weak augmentation alone as shown in appendix D We summarize the key differences between AugCL and prior work in Appendix Table 5.

### 4.3 SPLICE AUGMENTATION

Since AugCL is well suited to train with challenging strong augmentations, we introduce a novel augmentation called "**Splice**" to improve generalization in visual RL. RL generalization benchmarks that incorporate background distractions (Hansen & Wang, 2021; Stone et al., 2021) are challenging for state-of-the-art visual RL approaches. The standard solution is to introduce a variation of mix-up augmentation (Zhang et al., 2017) to RL training (Hansen & Wang, 2021; Fan et al., 2021; Hansen et al., 2021b). (Hansen & Wang, 2021) theorized that previous strategies failed to adapt to severe background distractions because task-relevant visual features such as the agent's shadow were removed.

Our new augmentation **Splice** solves this issue by pasting relevant visual features into an irrelevant background. This explicit separation of task-relevant versus task-irrelevant features helps generalization. Specifically, we mask out all non-relevant parts of the visual observation through a segmentation mask which is available in the simulation. We then replace all the non-task parts of the image with a random background image. We use COCO (Lin et al., 2014) for our experiments as the background replacement images. Further discussion and PyTorch (Paszke et al., 2019) style code for Splice and a visual example is provided in Appendix F.

## 5 EXPERIMENTS

We now evaluate AugCL and baselines on how well they can generalize to visual distribution shifts in the DMControl Generalization Benchmark (DMC-GB). In Section 5.1, we describe the experimental setup for how our method and baselines are configured. Next, in Section 5.2, we show that AugCL achieves state-of-the-art performance in the majority of settings in DMC-GB. Finally, in Section 5.3, we analyze what hyperparameters are necessary for the benefits of AugCL.

| Domain, Task | CURL | RAD | DrQ | PAD | SODA (conv) | SVEA (conv) | AugCL (conv) |
|---|---|---|---|---|---|---|---|
| Walker, Walk | $445 \pm 99$ | $400 \pm 61$ | $520 \pm 91$ | $468 \pm 47$ | $697 \pm 66$ | $760 \pm 145$ | $\mathbf{890 \pm 36}$ |
| Walker, Stand | $662 \pm 54$ | $644 \pm 88$ | $770 \pm 71$ | $797 \pm 46$ | $930 \pm 12$ | $942 \pm 26$ | $\mathbf{956 \pm 17}$ |
| Cartpole, Swingup | $454 \pm 110$ | $590 \pm 53$ | $586 \pm 52$ | $630 \pm 63$ | $831 \pm 21$ | $837 \pm 23$ | $\mathbf{852 \pm 9}$ |
| Ball In Cup, Catch | $231 \pm 92$ | $541 \pm 29$ | $365 \pm 210$ | $563 \pm 50$ | $892 \pm 37$ | $\mathbf{961 \pm 7}$ | $957 \pm 18$ |
| Finger, Spin | $691 \pm 12$ | $667 \pm 154$ | $776 \pm 134$ | $803 \pm 72$ | $901 \pm 51$ | $977 \pm 5$ | $\mathbf{980 \pm 9}$ |

Table 1: Results from DMC-GB benchmark color hard. All methods are evaluated on 5 seeds over 30 episodes. The mean and standard deviation are provided. AugCL outperforms baseline in 4 out the 5 tasks.

## 5.1 EXPERIMENTAL SETUP

**Environments and Evaluation**: The purpose of our experiments is to evaluate how well policies trained with various methods can generalize to new visual disturbances. All methods are first trained in a source environment without any visual disturbances. We then evaluate the trained policy in the same environment but with random visual disturbances. DMC-GB tests how methods can generalize to random colors, backgrounds, and camera poses. All methods are trained for 500,000 frames and evaluated on 5 tasks from DMC-GB in three different evaluation settings from DMC-GB (color-hard, video-easy, and video-hard). The 5 tasks from DMC-GB used in this paper are described in Appendix Table 8. We report the mean and standard deviation across 5 seeds per method, where each seed is evaluated by taking the average episode return across 30 episodes. For table 2 and table 3 we added training with SVEA during the weak augmentation phase.

**Baselines:** We compare AugCL against other recent pixel-based RL methods, some of which were explicitly designed for learning robust policies that can generalize to unseen environments. Specifically, we compare against **CURL**, **RAD**, **SVEA**, **SODA**, **DrQ** as well as **PAD** (Hansen et al., 2021a), which adapts to the test environment using self-supervision. Hyperparameters and baseline results are taken from (Hansen et al., 2021b). We don't compare to SECANT as it requires double the training frames to all other baselines and requires training 2 models sequentially. Also note that SVEA and SODA augment each batch twice, thus doubling the data the agent trains on whereas AugCL only uses a single batch.

**Data Augmentation Setup:** We apply random shift (Yarats et al., 2021) as our weak augmentation for AugCL. For all experiments, we selected $M = 200,000$ for AugCL, meaning we first perform 200k updates in the weak augmentation phase before switching to the strong augmentation phase. All hyperparameters shared between AugCL and baselines are kept the same. Random convolution produced the best results in prior works on color hard and overlay for video DMC-GB benchmarks (Hansen et al., 2021b), and we therefore use those augmentations for their respective benchmarks. Note that DrQ, AugCL and SVEA all use shift as their weak augmentation and CURL, PAD, RAD and SODA use crop. This is important to note as shift has been shown to give stronger empirical results than crop in DMC tasks (Yarats et al., 2021). "Overlay" in tables 2, 3 refers to (Hansen & Wang, 2021) version of mix-up. The original SVEA and SODA paper use the Places dataset (Zhou et al., 2018a) for Overlay, but during the time this paper was written Places was unavailable due to maintenance, so instead, we used COCO for AugCL. We felt this was a fair comparison as long as both datasets were different from RealEstate10k (Zhou et al., 2018b), which is used by DMC-GB. A full list of hyperparameters can be found in table 4. We apply random shift (Yarats et al., 2021) as our weak augmentation for AugCL and set $M = 200,000$, which we determined empirically. All overlapping hyper-parameters between methods are kept the same.

We also include results using Splice on the DMC-GB video easy and video hard benchmarks. In the DMControl tasks, we segment out the agent by filtering, converting the RGB image to HSV, and

| Domain, Task | CURL | RAD | DrQ | PAD | SODA (overlay) | SVEA (overlay) | AugCL (overlay) | SODA (splice) | SVEA (splice) | AugCL (splice) | AugCL + SVEA (splice) |
|---|---|---|---|---|---|---|---|---|---|---|---|
| Walker, Walk | $556 \pm 133$ | $600 \pm 63$ | $682 \pm 89$ | $717 \pm 79$ | $768 \pm 38$ | $819 \pm 71$ | $839 \pm 43$ | $625 \pm 29$ | $882 \pm 63$ | $879 \pm 35$ | $\mathbf{904 \pm 29}$ |
| Walker, Stand | $852 \pm 75$ | $745 \pm 146$ | $873 \pm 83$ | $935 \pm 20$ | $955 \pm 13$ | $961 \pm 8$ | $956 \pm 8$ | $875 \pm 71$ | $969 \pm 4$ | $958 \pm 7$ | $\mathbf{972 \pm 6}$ |
| Cartpole, Swingup | $404 \pm 67$ | $373 \pm 72$ | $485 \pm 105$ | $521 \pm 76$ | $758 \pm 62$ | $782 \pm 27$ | $795 \pm 12$ | $764 \pm 49$ | $850 \pm 32$ | $840 \pm 27$ | $\mathbf{854 \pm 9}$ |
| Ball In Cup, Catch | $316 \pm 119$ | $481 \pm 26$ | $318 \pm 157$ | $436 \pm 55$ | $875 \pm 56$ | $871 \pm 106$ | $910 \pm 34$ | $907 \pm 30$ | $963 \pm 11$ | $959 \pm 8$ | $\mathbf{967 \pm 3}$ |
| Finger, Spin | $502 \pm 19$ | $400 \pm 64$ | $533 \pm 119$ | $691 \pm 80$ | $695 \pm 97$ | $808 \pm 33$ | $782 \pm 40$ | $888 \pm 160$ | $975 \pm 20$ | $\mathbf{983 \pm 5}$ | $975 \pm 17$ |

Table 2: Results from DMC-GB benchmark video easy generalization benchmark. AugCL with the new splice augmentation outperforms prior work in 4 out of 5 of the tasks.

| Domain, Task | CURL | RAD | DrQ | PAD | SODA (overlay) | SVEA (overlay) | AugCL (overlay) | SODA (splice) | SVEA (splice) | AugCL (splice) | AugCL + SVEA (splice) |
|---|---|---|---|---|---|---|---|---|---|---|---|
| Walker, Walk | $58 \pm 18$ | $56 \pm 9$ | $104 \pm 22$ | $93 \pm 29$ | $381 \pm 72$ | $377 \pm 93$ | $364 \pm 65$ | $619 \pm 25$ | $861 \pm 59$ | $864 \pm 34$ | $\mathbf{888 \pm 30}$ |
| Walker, Stand | $45 \pm 5$ | $231 \pm 39$ | $289 \pm 49$ | $278 \pm 72$ | $711 \pm 83$ | $834 \pm 46$ | $610 \pm 127$ | $872 \pm 71$ | $960 \pm 6$ | $959 \pm 5$ | $\mathbf{962 \pm 7}$ |
| Cartpole, Swingup | $114 \pm 15$ | $110 \pm 16$ | $138 \pm 9$ | $123 \pm 24$ | $429 \pm 64$ | $393 \pm 45$ | $399 \pm 28$ | $594 \pm 90$ | $776 \pm 28$ | $742 \pm 35$ | $\mathbf{784 \pm 16}$ |
| Ball In Cup, Catch | $115 \pm 33$ | $97 \pm 29$ | $92 \pm 23$ | $66 \pm 61$ | $327 \pm 100$ | $403 \pm 174$ | $604 \pm 125$ | $750 \pm 260$ | $895 \pm 21$ | $\mathbf{916 \pm 21}$ | $905 \pm 35$ |
| Finger, Spin | $27 \pm 21$ | $32 \pm 11$ | $71 \pm 45$ | $56 \pm 18$ | $302 \pm 41$ | $335 \pm 58$ | $312 \pm 9$ | $873 \pm 163$ | $948 \pm 20$ | $952 \pm 24$ | $\mathbf{960 \pm 17}$ |

Table 3: Results from DMC-GB benchmark video hard. AugCL with the splice augmentation outperforms baselines in all of the tasks.

then taking pixels with HSV values only greater than a threshold. We set a consistent value threshold of 0.6 for all tasks to remove all aspects of the image, including the shadow, leaving only the agent. The hue threshold was 0 for all tasks except "Cartpole, Swingup" which required the hue threshold to be set to 3.5 due to the background being a mix of lighter and darker blues in these environments. The saturation threshold was set to 0 for all tasks. The full list of hyperparameters can be found in Appendix Table 4.

## 5.2 DMC-GB RESULTS

Firstly, Table 1 shows that AugCL outperforms all baselines in **4 out of 5** tasks in DMC-GB color-hard environments. This further closes the gap between the performance of the policy from training and its generalization performance on the test environment. Ball In Cup, Catch and Finger, Spin under color hard are close to matching the current SOTA in the train environment thanks to SVEA and AugCL as shown in Table 7. Despite SODA and SVEA using the same augmentations as AugCL and also being designed for generalization in pixel-based RL, AugCL outperforms them in evaluation return.

Next, in the DMC-GB video easy and video hard environments AugCL again outperforms baselines in almost all of the settings. AugCL outperforms baselines in **4 out of 5** tasks in video-easy (Table 2) and in **5 out of 5** tasks in video hard (Table 3). A combination of the splice augmentation and AugCL performs best. Splice combined with AugCL does well on "Finger, Spin" under video easy as it's only 1 average episodic reward off from the train environment SOTA as seen in Table 7. We theorize that Splice performs better than Overlay because Overlay is a weighted sum of pixels from the state image and an irrelevant image. (Laskin et al.; Cetin et al., 2022) showed the utility of weak augmentation was that a regularized CNN improved spatial attention mapping to task relevant features. Overlay may impede this process by making task relevant features less visible.

## 5.3 ABLATIONS

We analyze two design choices of AugCL: the curriculum and using separate critic networks for weak and strong augmentations. We use Random Convolution as the strong augmentation for all variations of AugCL in this section. **No Pretrain** in Figure 2 represents AugCL, but without the

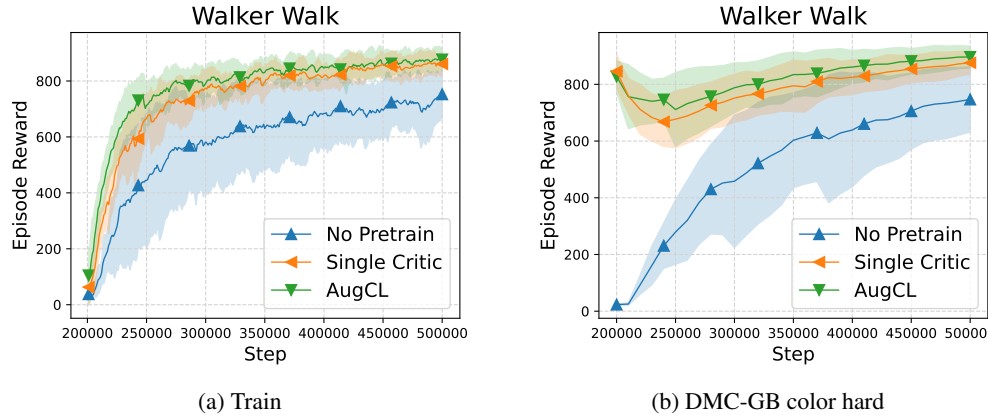

(a) Train

(b) DMC-GB color hard

Figure 2: **No Pretrain**: No weak augmented pre-training on the strong network. **Single Critic**: A single critic is used for training under weak and strong augmentation. The line represents the mean over 3 seeds, and the shadow represents variance. Figure 2a shows performance on the train environment, and 2b shows performance during training on DMC-GB color hard. The lines represent averages and the shaded regions the standard deviation of the results across 5 seeds.

copying of weights to the strongly augmented network at step $M$ (omitting line 9 in Alg. 1). **Single Critic**: represents having a single critic network for both strong and weak augmentation (omitting line 11 and line 17 becomes $\phi^{target} \leftarrow (1 - \zeta)\phi^{target} + \zeta\phi^S$ in Alg. 1).

As we can see without the pre-training even while bootstrapping from a Q-target network updated using a weakly augmented network. No Pretrain is much higher variance across the seeds and not able to match AugCL's performance on the walker walk task, thus showing the importance of first regularizing the CNN on weak augmentation, which motivates the curriculum. While Single Critic performs much better than No Pretrain, we can see it's much less sample efficient and converges to a lower solution than AugCL on both the train environment and the test environment. We can see that while Single Critic is improving the policy it is learning it is does not perform as well as distributing learning the state, action value to the weakly augmented network, thus showing the non-zero destabilization strong augmentation incurs.

We also experimented with setting $M = 0$ and found it was unable to learn as the strong critic couldn't learn a useful representation and could not bootstrap the target network's predictions to improve learning. We include further exploration of selecting $M$ in appendix A. This was indicated to us by the episodic reward not improving as training continued. We believe this also points towards the importance of weak augmentation regularization early in training. We believe that these experiments are clear evidence that strong augmentations do indeed incur a non-zero degradation to policy learning and that the key to getting good generalization performance is to disentangle the strong and weak augmentation as AugCL does, which is not possible without the curriculum as an unregularized CNN has difficulty learning task relevant representations.

## 6 CONCLUSION

AugCL shows tremendous improvements to generalized environments by disentangling strong and weak augmentations into their respective networks. The combination of AugCL and Splice have substantially improved performance on DMC-GB, giving a new SOTA. We also effectively show the importance of weak augmented pre-training for parallel weak and strong augmented network training, highlighting the missing ingredients to previous attempts. An issue with our method is selecting the optimal $M$ is still an open question. $M = 100,000$ yielded much worse results and we theorized that the CNN was not regularized enough. This tricky balance can lead to significant changes in results and we hope to find a more developed method for selecting $M$.

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

## A    Choice Of M On Performance

We explored how different M selections effect AugCL in figure 3. We found a parabolic relationship between M and average episodic reward. We see on the test environment the relationship between M and test environment performance is parabolic, with performance peaking at M=100k or 200k on the test environment. While it seemed that 100k and 200k for M gave the same performance higher and lower values of M had much lower average performance. We theorize this is due to striking a good balance between regularizing the CNN with weak augmentation and then training it to adapt to the strongly augmented version of the environment, which requires a lot of frames to approximate.

## B    DrQ vs RAD vs Non-naive Augmentation

In this paper we also explored the RAD style of augmentation Laskin et al. versus DrQ Yarats et al. (2021) and Non-Naive Augmentation Hansen et al. (2021b). In order to get the best generalization results possible we required achieving best results possible on the train environment as that acts as an upperbound to what can be achieved during generalization. We theorized the main strengths of DrQ stem from the shift augmentation introduced and the multiple sampling are for mitigating the issues introduced by RAD style augmentation. We show in Figure 5 that Non-Naive Augmentation does indeed perform better or equal to DrQ and RAD style augmentation in terms of average episodic reward on the train environment at the end of 500,000 train steps and is more sample efficient on **4 out of 5** tasks. RAD is equivalent to "DrQ 1k1m" in Figure 5. This is an exciting development as

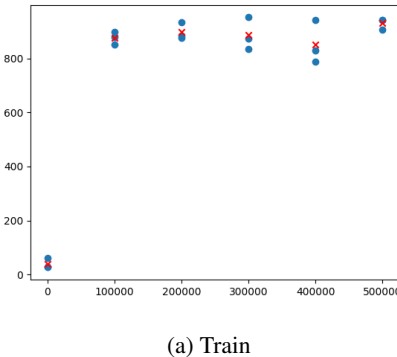
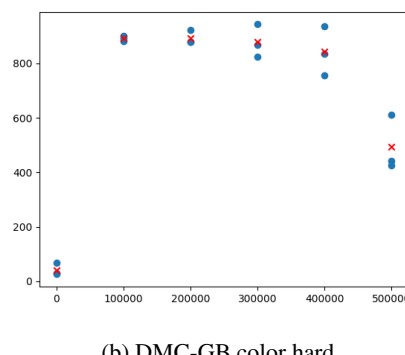

(a) Train

(b) DMC-GB color hard

Figure 3: We show the effects of M selection on AugCL by choosing [0, 100k, 200k, 300k, 400k, 500k]. AugCL is trained with random convolution as the strong augmentation on Walker, Walk for a total of 500k frames. Blue circles represent performance of different seeds using the average episodic reward over 50 runs and the red x represents the mean across all 3 seeds.

sampling multiple augmentations of images can be costly especially with off-policy methods which are known to take multiple days to train. Though more exploration is required as we only ran this assessment on 5 tasks in DMC, we encourage practitioners of pixel-based RL to leverage Non-Naive Augmentation over the current alternatives. Results shown in Figure 5.

## C NEURAL ARCHITECTURE

In this section we go into more detail about our neural architecture. Neural architecture are adopted from [Hansen et al. (2021b)]. A Convolutional Neural Network (CNN) encoder is shared between the actor and critic network. The encoder is only updated with respects to the Equation (1) and is detached during back propagation through Equation (3). The encoder is implemented with an 11-layer CNN encoder that takes a stack of RGB frames rendered at $84 \times 84 \times 3$ and output features of size $32 \times 21 \times 21$ where 32 is number of channels and $21 \times 21$ spatial feature mapping dimension. All convolutional layers use 32 filters and $3 \times 3$ kernels. The first convolutional layer has a stride of 2, while all others have a stride of 1. The CNN encoder is shared between the actor and critic and followed by linear projection layers for both the actor and critic. The linear projection layers consist of three fully connected layers with hidden dimension of 1024. All hyperparameters are shown in table 4.

## D AUGCL TRAIN ENVIRONMENT PERFORMANCE

We believe a key aspect of generalization includes maintaining the best policy possible on the train environment as well. We include in Table 6 train environment performance of AugCL across different strong augmentations we benchmarked on. We also include results for non-naive shift on the train environment as well as an upper bound to what all the generalized methods can achieve in Table 7.

## E TASKS DESCRIPTIONS

In this section we include a table with a brief description, dimensions of the action vector and whether the task is dense or sparse reward task in Table 8.

## F MORE DETAILS ON SPLICE AUGMENTATION

The inspiration for Splice came when we noticed that in many robotics task the relevant visual features had higher brightness. We noticed that DMC fit this criteria well as the ground and background

| Hyperparameter | Value |
|---|---|
| Frame Rendering | $3 \times 84 \times 84$ |
| Frames Stacked | 3 |
| Random Shift | 4 pixels |
| M | 200,000 |
| Action Repeat | 2 (finger), 8 (cartpole), 4 (otherwise) |
| Discount Factor $\gamma$ | 0.99 |
| Episode Length | 1000 |
| Learning Algorithm | SAC |
| Number Of Frames | 500,000 |
| Replay Buffer Size | 500,000 |
| Optimizer ($\beta$) | Adam($\beta_1$=0.9, $\beta_2$=0.999) |
| Optimizer ($\alpha$) | Adam($\beta_1$=0.5, $\beta_2$=0.999) |
| Learning Rate ($\theta$) | 1e-3 |
| Learning Rate ($\alpha$ of SAC) | 1e-4 |
| Batch Size | 128 |
| $\hat{\phi}$ Update Frequency | 2 |
| $\hat{\phi}$ Momentum Coefficient | 0.05(encoder), 0.01(critic) |
| Seeds | [0,4] |

Table 4: Hyperparameters used for all experiments

| Method | Weak Augmentation | Auxiliary Neural Architecture | Auxiliary loss | Modified Critic Loss | Uses Strong Augmentations |
|---|---|---|---|---|---|
| AugCL | Shift | Additional critic network | Additional Critic loss | N/A | Yes |
| SVEA | Shift | N/A | N/A | Data-mixing | Yes |
| SODA | Crop | Projection network | Contrastive loss | N/A | Yes |
| DrQ | Shift | N/A | N/A | Averaged Q & Q-target | No |
| CURL | Crop | N/A | InfoNCE | N/A | No |
| PAD | Crop | Self-supervised network | Adaptive self-supervision | N/A | No |

Table 5: This table summarizes what we assessed as key differences and similarities between methods. "N/A" stands for "Not Applicable."

tended to be a dark blue, while the agent is a combination of bright colors (typically yellow and a bright blue). Splice converts an RGB image to HSV color space then sets a threshold for hue, saturation and value. We use kornia [Riba et al. (2020)] for color space conversion. Hue represents color, saturation represents chromatic intensity and value represents brightness. If all values in a cell in the HSV converted image exceed the preset thresholds then they are imparted on a new image. In our case we splice out the agent and paste it onto a randomized background. Laskin et al.; Cetin et al. (2022) show weak augmentation leads to a better spatial attention mapping of features the agent can control like the robot over high frequency features like the background and flooring. Splice has the ability to impart human prior knowledge about the tasks through tuning the thresholds. By tuning the thresholds accordingly the user can parse out only the relevant visual features

| Strong Aug | Walker, Walk | Walker, Stand | Ball in Cup, Catch | Cartpole Swingup | Finger, Spin |
|---|---|---|---|---|---|
| Conv | $894 \pm 36$ | $958 \pm 13$ | $965 \pm 6$ | $853 \pm 6$ | $979 \pm 9$ |
| Overlay | $903 \pm 26$ | $969 \pm 7$ | $964 \pm 5$ | $869 \pm 10$ | $967 \pm 9$ |
| Splice | $878 \pm 38$ | $958 \pm 7$ | $962 \pm 6$ | $865 \pm 13$ | $976 \pm 15$ |

Table 6: Train environment performance at the end of 500,000 train steps of AugCL with varying strong augmentations. The mean and standard deviation over 5 seeds where each seed is evaluated using the mean of 30 episodes done for each seed.

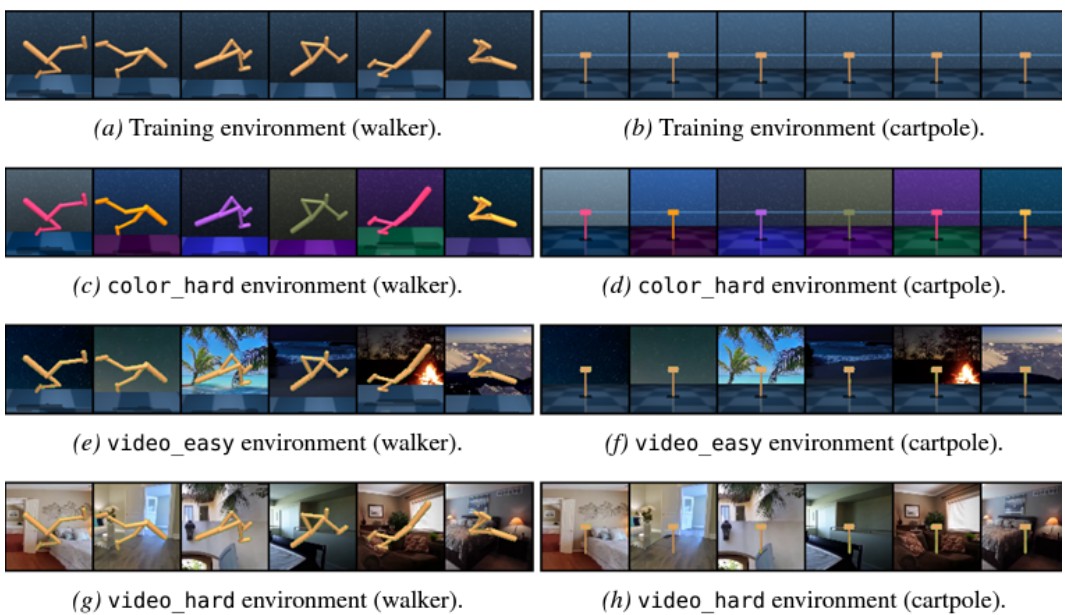

Figure 4: Image taken from [Hansen et al. (2021b)] with examples from cartpole and walker of DMC-GB.

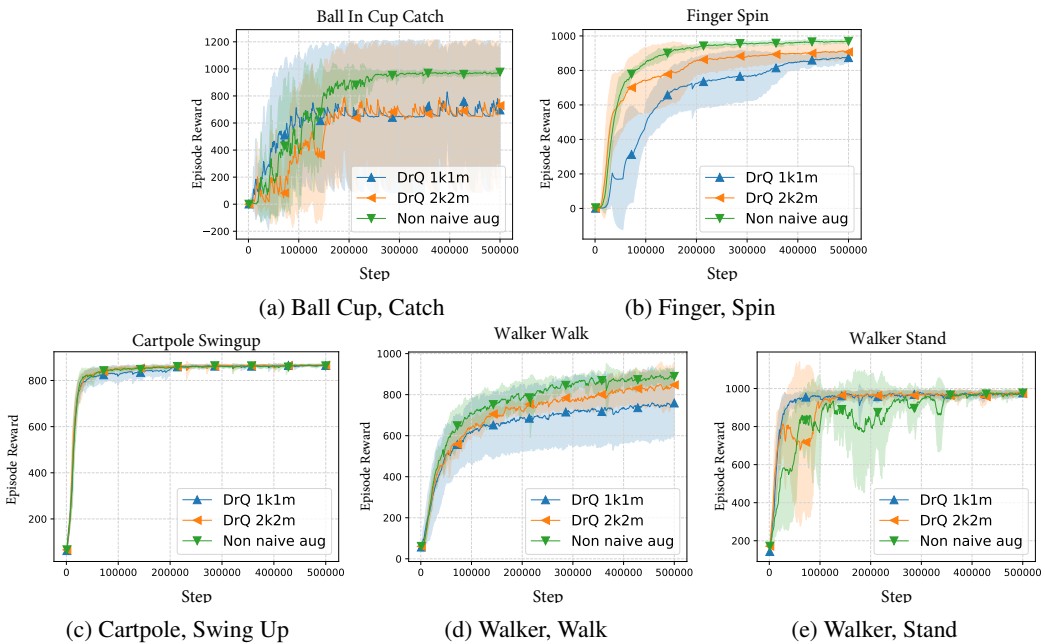

Figure 5: We compare DrQ and Non-Naive Augmentation on the train environment. DrQ with k, m=1 and k, m=2 where $k$ represents number of augmented samples for $s_t$ in Equation (1) and $m$ represents number of augmented samples for $s_{t+1}$, all methods use shift. We find that non-naive augmentation with shift is more sample efficient and lower performance variance between seeds on 4 out of 5 tasks. The mean and standard deviation of the average of 30 episodes across 3 seeds: [0, 2] are used to generate plots. The lines represent the means and the shadow represents the variance.

in the task. This allows us to circumvent the high frequency feature bias that CNNs inherently have by pasting the relevant features on random backgrounds, thus the high frequency features between

| Walker, Walk | Walker, Stand | Ball in Cup, Catch | Cartpole Swingup | Finger, Spin |
|---|---|---|---|---|
| $916 \pm 25$ | $975 \pm 2$ | $971 \pm 5$ | $869 \pm 11$ | $984 \pm 2$ |

Table 7: Shift augmentation evaluation results on the train environment across 5 seeds with the mean and standard deviation across 30 episodes for each seed. This table serves as an upper bound to what can be achieved in generalized benchmarks.

| Domain, Task | Description | Action Vector Size | Dense Rewards |
|---|---|---|---|
| Walker, Walk | A planar walker that is rewarded for walking forward at a target velocity. | 6 | Yes |
| Walker, Stand | A planar walker that is rewarded for standing with an upright torso at a constant minimum height. | 6 | Yes |
| Cartpole, Swingup | Swing up and balance an unactuated pole by applying forces to a cart at its base. The agent is rewarded for balancing the pole within a fixed threshold angle. | 1 | Yes |
| Ball In Cup, Catch | An actuated planar receptacle is to swing and catch a ball attached by a string to its bottom | 2 | No |
| Finger, Spin | A manipulation problem with a planar 3 DoF finger. The task is to continually spin a free body | 2 | No |

Table 8: Table containing: action space dimension, brief description of task and if rewards are dense. Descriptions are taken from Hansen et al. (2021b).

frames becomes the task relevant features. Example code is given below an a comparative example is shown in Figure 6.

```python
import torch
import kornia

def splice(x, hue_t, saturation_t, value_t):
    b, _, h, w = x.shape
    x_HSV = kornia.color.rgb_to_hsv(x)
    overlay = sample_background(batch_size=b)
    thresholds = torch.FloatTensor([hue_t, saturation_t, value_t])
    thresholds = thresholds.view(1, -1, 1, 1).repeat(b, 1, h, w)
    mask = x_HSV > thresholds
    mask = torch.all(mask, dim=1)
    overlay[mask] = x[mask]
    return overlay
```

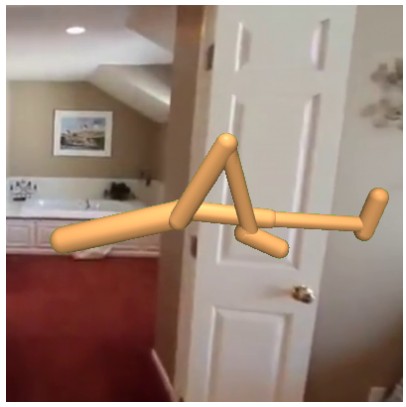
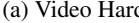

(a) Video Hard

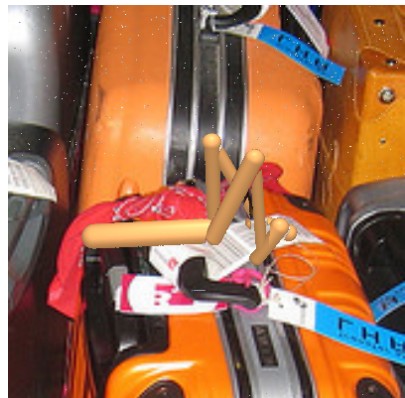

(b) Splice

Figure 6: Value threshold of 0.6, with threshold of 0 set for hue and saturation for splice augmentation shown in Figure 6b. Figure 6a taken from DMC-GB.

