# OpenReview forum: "Augmentation Curriculum Learning For Generalization in RL"
_ICLR.cc/2023/Conference — Submitted to ICLR 2023_

### Official Review · Reviewer_QnFd · 2022-10-25

**Confidence:** 3
**Correctness:** 3
**Technical Novelty And Significance:** 2
**Empirical Novelty And Significance:** 2
**Recommendation:** 5

**Clarity, Quality, Novelty And Reproducibility:**

The proposed method is novel, but its impact is questionable, which reduces the overall quality of the paper.
The authors have provided their code with instructions, to be used in combination with the DMC benchmark repository, which significantly contributes to reproducibility.
The paper is written in a very clear and concise manner.

**Strength And Weaknesses:**

> Strengths

The paper is well-written, well-structured and easy to follow.
The paper achieves SOTA performance on 13/15 environments of a popular benchmark compared to previous methods. The authors prove the relevance of the key aspects of their proposed methods by running an ablation study.

> Weaknesses

The authors conclude the value of M (timesteps for the1st augemntation phase) is an open question. I believe that the determining of this value is rather crucial, as the whole notion of curriculum (and hence the core method) relies on this parameter. The authors do report experimenting with 0 and 100,000 as values for M, which yielded poor results. I would like to see more analysis on how to select M and what effect this value beholds in other environments.
The paper talks about addressing how to effectively incorporate strong augmentations such as random convolutions and overlays, but then mainly focuses on their own proposed augmentation (Splice) in their experiments. Results using the Overlay augmentation are indeed presented, but 1) originate from a different dataset than which was used in SVEA and SODA, and 2) are not superior in all environments compared to previous works. It can thus be argued, that the proposed method AugCL does not necessarily surpass previous methods, but the new augmentation method (Splice) is the reason for improved performance. It would be interesting to see how much the proposed method outperforms previous methods using the same augmentations.
The contribution of the new augmentation method Splice is marginal, as it is not applicable in most other RL benchmarks. Since Splice is tailored for the DMC suite, it cannot be considered as a general augmentation method, which is why I would consider it a contribution of low significancy.

**Summary Of The Paper:**

The paper presents AugCL, a method for learning to generalize to unseen environments in pixel- based RL using strong visual augmentations. Training is split into two phases of using weak and strong augmentations respectively to leverage a curriculum. The authors use a separate critic network for each phase and demonstrate how this disentangelment induces superior generalization. They further introduce an image augmentation method called Splice, and show how the combination of this with their method achieves state-of-the-art performance on the DMC-GB benchmark.

**Summary Of The Review:**

The paper utilises a very trivial method to marginally improve generalization of pixel-based RL on one benchmark. The contribution is novel, however, it is not empirically proved to be beneficial in other benchmarks, due to which I don't find it to be particularly significant. Further, it is dubious to label the simplistic two-phased learning as a curriculum.

---

> ### Author Response · Authors · 2022-11-18
> **Addressing The Generalization Of Splice**
>
> 1) Since Splice is tailored for the DMC suite, it cannot be considered as a general augmentation method, which is why I would consider it a contribution of low significancy.
>
> We disagree that Splice is tailored to DMC only. Splice can filter pixels by brightness, colour or saturation. It can be applied to some Robosuite[1] tasks where the agent is a brighter white and the background is a darker blue. As well as many Atari[2] games where the background is black and the relevant features are brighter colours. Car2Racing[2] also has a green background while the agent and road are differing colours, since Splice can filter by colour the relevant features (road and car) can be parsed out. A segmentation mask can also be made in Habitat 2.0[3] and used to splice out task-relevant features as well. Segmentation masking or other simulator properties can be spliced out using this method. Other than distracting backgrounds we also experimented with applying colour changes with Splice. Since augmentations such as random jitter or random convolution apply a deterministic function to an image indiscriminately this might not be desirable in environments such as DMC-GB[4] where the agent, background and flooring have their colours randomly changed independently of each other. Splice would allow us to apply colour changes independently to different features in an image.
>
> 2) The authors do report experimenting with 0 and 100,000 as values for M, which yielded poor results. I would like to see more analysis on how to select M and what effect this value beholds in other environments.
>
> We have included the experiments on the selection of M in our paper and a table here. The table provides the mean performance over 3 seeds where each model is averaged over 30 episodes at differing Ms:
>
> | Environment | M=0 | M=100k | M=200k | M=300k | M=400k | M=500k |
> |-------------|-----|--------|--------|--------|--------|--------|
> | Train       | 39  | 877    | 898    | 887    | 852    | 930    |
> | Test        | 34  | 892    | 892    | 878    | 842    | 464    |
>
> 3) not superior in all environments compared to previous works. It can thus be argued, that the proposed method AugCL does not necessarily surpass previous methods, but the new augmentation method (Splice) is the reason for improved performance. It would be interesting to see how much the proposed method outperforms previous methods using the same augmentations.
>
> We have included the results you request. We trained SODA and SVEA with Splice
>
> $\textbf{DMC GB video easy:}$
> | Domain, Task       | SODA      | SVEA     | AugCL       | AugCL + SVEA |
> |--------------------|-----------|----------|-------------|--------------|
> | Walker, Walk       | 625 ± 29  | 882 ±63  | 879 ±35     | **904 ± 29** |
> | Walker, Stand      | 875 ± 71  | 969 ±4   | 958 ± 7     | **972 ± 6**  |
> | Cartpole, Swingup  | 764 ± 49  | 850 ± 32 | 840 ±27     | **854 ± 9**  |
> | Ball In Cup, Catch | 907 ±30   | 963 ± 11 | 959 ± 8     | **967 ± 3**  |
> | Finger, Spin       | 888 ± 160 | 975 ± 20 | **983 ± 5** | 975 ± 17     |
>
> $\textbf{DMC GB video hard:}$
> | Domain, Task       | SODA      | SVEA     | AugCL        | AugCL + SVEA |
> |--------------------|-----------|----------|--------------|--------------|
> | Walker, Walk       | 619 ± 25  | 861 ± 59 | 864 ± 34     | **888 ± 30** |
> | Walker, Stand      | 872 ± 71  | 960 ± 6  | 959 ± 5      | **962 ± 7**  |
> | Cartpole, Swingup  | 594 ± 90  | 776 ± 28 | 742 ± 35     | **784 ± 16** |
> | Ball In Cup, Catch | 750 ± 260 | 895 ± 21 | **916 ± 21** | 905 ± 35     |
> | Finger, Spin       | 873 ± 163 | 948 ± 20 | 952 ± 24     | **960 ± 17** |

---

> > ### Comment · Reviewer_QnFd · 2022-12-07
> > **Thank you.**
> >
> > I will keep my score. I still think selecting M is not convincing as a CL algorithm.

---

### Official Review · Reviewer_iZtt · 2022-10-25

**Confidence:** 3
**Correctness:** 3
**Technical Novelty And Significance:** 2
**Empirical Novelty And Significance:** 2
**Recommendation:** 6

**Clarity, Quality, Novelty And Reproducibility:**

The paper is fairly clear, but its novelty is somewhat limited. Perhaps I am lacking context here but if I understand correctly training two separate critics for different augmentations is not new, although this particular setup may be an improvement over past work. In-pasting natural image distractors is also not new, although the specifics here may matter for performance.

I believe I could reproduce this paper or at least its broad strokes relatively easily.

Note on formatting: it seems all the non-inline citations are missing parentheses. Are you using the `\citep{...}` command?


**Strength And Weaknesses:**

The paper is relatively clear and well motivated. It introduces methods that make sense and motivates them well.

I have some issues with the empirical evaluation of the proposed methods:
- It seems that one of the the main drivers of performance wrt baselines is the augmentation used. Were any baselines trained with Splice?
- As the authors point out, the role of $M$ is still unclear, but its role is a central part of the method.
    - Here's a suggestion for a plot: run experiments with varying $M$s and display results with $M$ on the $x$ axis and the AUC or average reward after training on the $y$ axis for two settings, in-distribution and out-of-distribution MDPs.
    - If what the authors posit is correct, with $M=0$ training should be too hard and not perform well on either MDPs, while with $M\to \infty$ (or simply $M>$total number of steps trained on) training should not generalize well to OOD MDPs but work well on in-distribution MDPs; and there should be a sweet spot in the middle where performance is ideal. Not only would this be a nice ablation, it would provide evidence that the _hypotheses_ that underlie this paper are correct.

Other than that, I'm also a bit unsure that invoking curriculum learning is correct here. While you could argue this is _technically_ CL, with two phases we typically only say the the first phase is pretraining. A curriculum on the other hand usually implies a continuously changing "difficulty level" and data distribution adjustment, and one that is _adaptive_ to the learner's performance--here $M$ is a hyperparameter.

**Summary Of The Paper:**


This paper proposes to decouple critics in an actor-critic scenario, where one critic is trained on so-called weak pixel-based augmentations and the other on strong augmentations. The argument is that while weak augmentations help stabilize training within an MDP, strong augmentations help generalize to a larger set of variations on that MDP but destabilize training.

- This induces four parameterized functions, $\pi_\psi$, $Q_{\phi^W}$, $Q_{\phi^{target}}$, and $Q_{\phi^S}$ ($W$ and $S$ standing for weak and strong).
- A new strong augmentation is introduced, called Splice, which pastes in natural image distractors forcing the model to pay attention to the relevant parts of the image.
- Training is separated into two phases. The method introduces a hyperparameter $M$, the number of steps for which training is in the first phase.
  - For $M$ steps, $\pi_\psi$ is trained on $Q_{\phi^W}$, $Q_{\phi^W}$ is bootstrapped to the EMA (of $\phi^W$) $Q_{\phi^{target}}$ with weak augmentations for both $\pi$ and $Q$
  - Then, $\pi_\psi$ is trained on $Q_{\phi^S}$ with strong augmentations, $Q_{\phi^W}$ and $Q_{\phi^S}$ are bootstrapped to the EMA (of $\phi^W$) $Q_{\phi^{target}}$ with weak and strong augmentations respectively.
- This method shows improvements on an array of environments from the DeepMind control benchmarks.


**Summary Of The Review:**

While improvements in performance are always valuable, the scientific contribution here appears relatively minor to me. The proposed methods build on existing ones, but without deeply investigating them. This leaves us with higher numbers but not much new knowledge.

As is I'm on the fence about accepting this paper, but I think it could be a much stronger contribution with a proper empirical validation _of the hypotheses_.

Update: the new results provide more clarity to this paper. In particular they test the hypothesis that $M$ matters, this is interesting in itself, but the other interesting aspect is that in terms of performance, it is the Splice augmentation that seems to offer most of the boost. I'm still leaning towards accept.

---

> ### Author Response · Authors · 2022-11-18
> **Analysis Of Selection Of M**
>
> We would like to thank this reviewer for pointing out formatting concerns and are delighted to let them know that all suggestions have been addressed appropriately.
> 1) As the authors point out, the role of  is still unclear, but its role is a central part of the method.
> Here's a suggestion for a plot: run experiments with varying s and display results with  on the  axis and the AUC or average reward after training on the  axis for two settings, in-distribution and out-of-distribution MDPs.
> If what the authors posit is correct, with  training should be too hard and not perform well on either MDPs, while with  (or simply total number of steps trained on) training should not generalize well to OOD MDPs but work well on in-distribution MDPs; and there should be a sweet spot in the middle where performance is ideal. Not only would this be a nice ablation, it would provide evidence that the hypotheses that underlie this paper are correct.
>
> We have also included the requested experiments on M=0 to M=infinity here. The table provides the mean performance over 3 seeds at differing Ms. We chose DMC-GB color hard setting and used random convolution as our strong augmentation. We can see the relationship between M and test environment performance is parabolic. The sweet spot seems to be between 100k and 200k, but if we want optimal performance on both the train and test environment we can see 200k is ideal. The issue with higher values of M is that the agent may not see enough frames under strong augmentation in order to get a good approximation of the distribution shift. This must be delicately balanced with ensuring the agent is regularized enough as well via weak augmentation.
>
> | Environment | M=0 | M=100k | M=200k | M=300k | M=400k | M=500k |
> |-------------|-----|--------|--------|--------|--------|--------|
> | Train       | 39  | 877    | 898    | 887    | 852    | 930    |
> | Test        | 34  | 892    | 892    | 878    | 842    | 464    |
>
>
>
> 2) It seems that one of the the main drivers of performance wrt baselines is the augmentation used. Were any baselines trained with Splice?
>
> $\textbf{DMC GB video easy:}$
> | Domain, Task       | SODA      | SVEA     | AugCL       | AugCL + SVEA |
> |--------------------|-----------|----------|-------------|--------------|
> | Walker, Walk       | 625 ± 29  | 882 ±63  | 879 ±35     | **904 ± 29** |
> | Walker, Stand      | 875 ± 71  | 969 ±4   | 958 ± 7     | **972 ± 6**  |
> | Cartpole, Swingup  | 764 ± 49  | 850 ± 32 | 840 ±27     | **854 ± 9**  |
> | Ball In Cup, Catch | 907 ±30   | 963 ± 11 | 959 ± 8     | **967 ± 3**  |
> | Finger, Spin       | 888 ± 160 | 975 ± 20 | **983 ± 5** | 975 ± 17     |
>
> $\textbf{DMC GB video hard:}$
> | Domain, Task       | SODA      | SVEA     | AugCL        | AugCL + SVEA |
> |--------------------|-----------|----------|--------------|--------------|
> | Walker, Walk       | 619 ± 25  | 861 ± 59 | 864 ± 34     | **888 ± 30** |
> | Walker, Stand      | 872 ± 71  | 960 ± 6  | 959 ± 5      | **962 ± 7**  |
> | Cartpole, Swingup  | 594 ± 90  | 776 ± 28 | 742 ± 35     | **784 ± 16** |
> | Ball In Cup, Catch | 750 ± 260 | 895 ± 21 | **916 ± 21** | 905 ± 35     |
> | Finger, Spin       | 873 ± 163 | 948 ± 20 | 952 ± 24     | **960 ± 17** |

---

> > ### Comment · Reviewer_iZtt · 2022-11-18
> > **Re: Analysis**
> >
> > Thanks for the precisions and further experiment results.

---

### Official Review · Reviewer_3K31 · 2022-10-26

**Confidence:** 2
**Correctness:** 3
**Technical Novelty And Significance:** 2
**Empirical Novelty And Significance:** 2
**Recommendation:** 5

**Clarity, Quality, Novelty And Reproducibility:**

I think given that the method in this paper beats baselines handily on the DMCGB, it's definitely an acceptable quality method. Tables 2 and 3 suggest to me that splice augmentation is the main reason that the method won the benchmark, not the particular choice of losses that the authors use for incorporating splice augmentation. Including SODA and SVEA with splice augmentation in the tables would go a long way in clarifying the importance of each of the authors' contributions. With respect to originality, the method seems very similar to SVEA to me.

**Strength And Weaknesses:**

The main strength of this paper is the empirical performance. Achieving the top performance on the DMCGB is certainly important. The method proposed in the paper beats baselines in all but 1 of the DMCGB tasks.

However, I do think there are a number of ways the paper needs to improve. Currently, I think the paper feels a bit rushed and a few edits and ablations would increase the impact of the paper substantially. I've included a list of comments and edits below.

- the authors are using the ICLR 2022 template

- Section 3.1 should just be $R$? not $R_t$

- Algorithm 1 - in lines 13 and 16, $L_\pi$ should take as input a $\phi$ as well to specify which Q function weights are being used in the equations in section 4.1 (I think in line 13, it should be $\phi^S$ and in line 16 it should be $\phi^W$?). I think this detail is important for understanding the algorithm.

- add equation labels for the equations in section 4.1

- at the top of page 5, I think the authors should reference a line different from "line 1"?

- I don't think Figure 1 is particularly informative since the arrows don't communicate which parts of the loss each model is being used for. I would personally prefer if the authors wrote out explicitly the loss for the policy and critic(s) in phase 1 and phase 2. For instance, the weak critic and target critic are updated the same way in phase 1 and 2, the only thing that changes in phase 2 is we begin training the strong critic and the policy uses the strong critic in its update (I believe, although currently ambiguous in line 13 of Algorithm 1). Currently, it's not obvious to me looking at Figure 1 that the updates for two of the networks are unchanged in phase 2, but that information would help a lot for internalizing the author's method.

- Table 2 and Table 3 - did the authors evaluate SODA and SVEA with splice? The authors should demonstrate that SODA and SVEA don't get the same performance boost as AugCL when switching from overlay to splice. Otherwise my sense is the main contribution of the paper is the use of splice augmentation, not the particular choice of training weak and strong critics in parallel which is what the authors emphasize in the method section.

- page 9 - ideally the authors would sweep across 3-5 values of M to get a sense of how sensitive the algorithm is to choice of M

**Summary Of The Paper:**

This paper tackles the problem of generalization in reinforcement learning. The authors seek to find algorithms for training a policy such that the policy will perform well on novel test-time environments. The authors propose a two-phase approach. In phase 1, the policy and value function are both trained with "weak" data augmentation. In phase 2, a second value function is trained in parallel with "strong" data augmentation and is used to update the policy. The method achieves SOTA performance on the Deep Mind Control Generalization Benchmark (DMCGB).

**Summary Of The Review:**

Currently, I feel this paper is somewhat difficult to read and the method has limited impact outside of the DMCGB. However, if the authors address my comments in the "Strengths and Weaknesses" section, I will consider increasing my score.

---

> ### Author Response · Authors · 2022-11-18
> **Sweep over M Experiments**
>
> We really appreciate your time and detailed feedback on how to improve the writing and understandability of this paper! We have incorporated all the feedback you have given.
>
> 1) Ideally the authors would sweep across 3-5 values of M to get a sense of how sensitive the algorithm is to choice of M
>
> We have included a sweep over M in our paper and a table here. The table provides the mean performance over 3 seeds at differing Ms. We chose DMC-GB color hard setting and used random convolution as our strong augmentation. We can see the relationship between M and test environment performance is parabolic. The sweet spot seems to be between 100k and 200k, but if we want optimal performance on both the train and test environment we can see 200k is ideal. The issue with higher values of M is that the agent may not see enough frames under strong augmentation in order to get a good approximation of the distribution shift. This must be delicately balanced with ensuring the agent is regularized enough as well via weak augmentation.
>
> | Environment | M=0 | M=100k | M=200k | M=300k | M=400k | M=500k |
> |-------------|-----|--------|--------|--------|--------|--------|
> | Train       | 39  | 877    | 898    | 887    | 852    | 930    |
> | Test        | 34  | 892    | 892    | 878    | 842    | 464    |
>
>
> 2) Table 2 and Table 3 - did the authors evaluate SODA and SVEA with splice? The authors should demonstrate that SODA and SVEA don't get the same performance boost as AugCL when switching from overlay to splice. Otherwise my sense is the main contribution of the paper is the use of splice augmentation, not the particular choice of training weak and strong critics in parallel which is what the authors emphasize in the method section
>
> We have included splice results from SODA and SVEA
>
> $\textbf{DMC GB video easy:}$
> | Domain, Task       | SODA      | SVEA     | AugCL       | AugCL + SVEA |
> |--------------------|-----------|----------|-------------|--------------|
> | Walker, Walk       | 625 ± 29  | 882 ±63  | 879 ±35     | **904 ± 29** |
> | Walker, Stand      | 875 ± 71  | 969 ±4   | 958 ± 7     | **972 ± 6**  |
> | Cartpole, Swingup  | 764 ± 49  | 850 ± 32 | 840 ±27     | **854 ± 9**  |
> | Ball In Cup, Catch | 907 ±30   | 963 ± 11 | 959 ± 8     | **967 ± 3**  |
> | Finger, Spin       | 888 ± 160 | 975 ± 20 | **983 ± 5** | 975 ± 17     |
>
> $\textbf{DMC GB video hard:}$
> | Domain, Task       | SODA      | SVEA     | AugCL        | AugCL + SVEA |
> |--------------------|-----------|----------|--------------|--------------|
> | Walker, Walk       | 619 ± 25  | 861 ± 59 | 864 ± 34     | **888 ± 30** |
> | Walker, Stand      | 872 ± 71  | 960 ± 6  | 959 ± 5      | **962 ± 7**  |
> | Cartpole, Swingup  | 594 ± 90  | 776 ± 28 | 742 ± 35     | **784 ± 16** |
> | Ball In Cup, Catch | 750 ± 260 | 895 ± 21 | **916 ± 21** | 905 ± 35     |
> | Finger, Spin       | 873 ± 163 | 948 ± 20 | 952 ± 24     | **960 ± 17** |

---

### Official Review · Reviewer_wrjJ · 2022-10-28

**Confidence:** 4
**Correctness:** 4
**Technical Novelty And Significance:** 2
**Empirical Novelty And Significance:** 2
**Recommendation:** 3

**Clarity, Quality, Novelty And Reproducibility:**

I suggest the authors double check that citations are performed correctly according to the ICLR style guidelines - reading the citations throughout the paper was jarring and disorienting due to the lack of differentiation within the sentences.

I can identify no further issue with the clarity, quality, or reproducibility of this work.  In terms of the novelty, I believe the work is slightly novel; I believe that exploring a mix of strong and weak augmentations exists in prior art and the novelty of this work mostly revolves around this two-stage curriculum schedule.  In terms of the Splice data augmentation, I believe similar works have been done before [1, 2], where the agent and task-relevant features are extracted out and placed over a different distracting background.  There were no model or architectural improvements - the authors apply their curriculum on top of an existing SAC model.

[1] Stone et al. “The Distracting Control Suite – A Challenging Benchmark for Reinforcement Learning from Pixels”, 2021.

[2] Hansen et al. “Stabilizing Deep Q-Learning with ConvNets and Vision Transformers under Data Augmentation”, 2021.

**Strength And Weaknesses:**

A strength of this paper is in its simple, easy to grasp idea - this would definitely help it be applied out-of-the box to a variety of RL methods.

A main weakness of this paper is in its novelty, which will be elucidated upon in the section below.  Furthermore, I sense that a major weakness of this paper is how restricted it is to the DMC task suite.  Firstly, the selection of weak and strong augmentations were predefined specifically for the DMC tasks (‘Known weak augmentations for all DMC tasks are: crop, translate, and shift’).  This limits the application of the proposed model to other domains for which there is no prior knowledge about the weakness or strength of augmentations.  It would be interesting if the authors could automatically uncover such properties of elements in a proposed augmentation set automatically - however, as such, it looks like this approach was specifically designed for the DMC task suite rather than being general.

Also, the numerical results of the proposed AugCL do not appear to outperform the existing art (in particular, SVEA) on the listed tasks, with the exception of Walker, Walk.  Given how design decisions were specifically selected for this domain, I would have expected more substantial performance improvements.

The claim that AugCL enables generalization to unseen environments would also be greatly reinforced if it were demonstrated across true unseen visual domains and tasks, rather than remaining in the DeepMind Control suite.  I believe that given the claims of the paper, a more thorough set of experiments would be useful in demonstrating the power of their proposed curriculum.

**Summary Of The Paper:**

In this work, the authors tackle the problem of generalization to novel visual environments in Reinforcement Learning.  The authors differentiate between weak augmentations (those that help with learning to solve the task in the same environment) and strong augmentations (those that hurt the training performance on the same environment, but may help generalization across domains).  The authors propose a curriculum where after M steps of only training on the weak augmentations, additional optimization on strong augmentations is included as well.  The authors also propose a data augmentation called Splice, where task-relevant information is cut out from the environment and pasted over a distracting background.

**Summary Of The Review:**

The proposed approach in this work is intuitive, and simple to apply to many current RL solutions.  However, I believe it suffers from explicit domain-specific information (distinguishing weak from strong augmentations specifically for DMC), as well as a lack of convincing experiments that fully showcase the strength of the model with respect to the claims.  I also believe the only substantial novel contribution in this work is the curriculum schedule (as the differentiation of weak vs strong augmentations is assumed, not learned/identified ad hoc per environment), which is limited in terms of its intellectual impact.  I therefore recommend that this paper undergo continued revisions before being considered for acceptance.

---

> ### Author Response · Authors · 2022-11-18
> **Addressing Generalization Concerns**
>
> We would like to thank this reviewer for their feedback. We have fixed the paper to match ICLR 2023 style guide as mentioned by the reviewer.
>
> 1) I sense that a major weakness of this paper is how restricted it is to the DMC task suite. Firstly, the selection of weak and strong augmentations were predefined specifically for the DMC tasks (‘Known weak augmentations for all DMC tasks are: crop, translate, and shift’).
>
> The image augmentations Shift and Crop are generally accepted as weak augmentations (augmentations that improve average episodic reward on the training environment) across a variety of pixel-based tasks beyond just DMC (iGibson[1], Robosuite[1][5], DMC[1][2][3], Atari[3] and CARLA[1]). We only wrote: “known weak augmentations for all DMC tasks are: crop, translate, and shift” in our paper in section 4.1 in the second to last paragraph is because though shift and crop are generally considered weak augmentations they have only been shown to be a weak augmentation for a couple of tasks in Procgen, while have been detrimental to policy learning for a couple of the other tasks in Procgen.
>
> 2) This limits the application of the proposed model to other domains for which there is no prior knowledge about the weakness or strength of augmentations.
>
> In most tasks, it’s generally safe to assume that shift or crop will help policy learning as shown in [3] that it has a regularisation effect on the CNN to stop it from overfitting to its own Q-value estimates. In terms of strong augmentation, random convolution and mix-up have not been shown to improve train environment performance on any environment as far as we know and thus are generally considered strong augmentations. Though mix-up and random convolution are strong augmentations their utility lies in their visual simulator to difficult visual distribution shifts that are important to build robustness to like changes of color or distracting backgrounds. The key point isn’t that you’re selecting a strong augmentation, it’s that you’re selecting an augmentation reflective of the potential distribution shifts in deployment you will see. Therefore it can generally be applied by choosing shift as its weak augmentation and selecting strong augmentations based on their visual similarity to what you anticipate will be seen in deployment.
>
> 3) The claim that AugCL enables generalization to unseen environments would also be greatly reinforced if it were demonstrated across true unseen visual domains and tasks, rather than remaining in the DeepMind Control suite.
>
> Augmentations haven’t shown any empirical evidence in improving robustness to unseen tasks as far as we know, only in building robustness to visual perturbations on the task trained under, which was the aim of this paper. The methods we benchmarked against [2][5][7], used DMC as their primary way of evaluating their methods as well and they haven’t shown any issue in terms of generalization.
>
> 4) In terms of the Splice data augmentation, I believe similar works have been done before [1, 2].
>
> [5] and [6] the two papers you have referenced incorporate distracting backgrounds by modifying the simulator itself. Splice is the first and only way this is achieved through image augmentation.
>
>
>
> $\textbf{Works Cited}$:
>
> [1] Linxi Fan, Guanzhi Wang, De-An Huang, Zhiding Yu, Li Fei-Fei, Yuke Zhu, and Animashree Anandkumar. Secant: Self-expert cloning for zero-shot generalization of visual policies. In Marina Meila and Tong Zhang (eds.), Proceedings of the 38th International Conference on Machine Learning, volume 139 of Proceedings of Machine Learning Research, pp. 3088–3099. PMLR,
> 18–24 Jul 2021. URL https://proceedings.mlr.press/v139/fan21c.html.
>
> [2] Denis Yarats, Ilya Kostrikov, and Rob Fergus. Image augmentation is all you need: Regularizing deep reinforcement learning from pixels. In International Conference on Learning Representations, 2021. URL https://openreview.net/forum?id=GY6-6sTvGaf
>
> [3] Edoardo Cetin, Philip J Ball, Stephen Roberts, and Oya Celiktutan. Stabilizing off-policy deep reinforcement learning from pixels. In International Conference on Machine Learning, pp. 2784–2810. PMLR, 2022.
>
> [4] Michael Laskin, Kimin Lee, Adam Stooke, Lerrel Pinto, Pieter Abbeel, and Aravind Srinivas. Reinforcement learning with augmented data. arXiv:2004.14990.
>
> [5] Nicklas Hansen and Xiaolong Wang. Generalization in reinforcement learning by soft data augmentation. In International Conference on Robotics and Automation, 2021.
>
> [6] Austin Stone, Oscar Ramirez, Kurt Konolige, and Rico Jonschkowski. The distracting control suite – a challenging benchmark for reinforcement learning from pixels. arXiv preprint arXiv:2101.02722, 2021.
>
> [7] Hansen, N., Su, H., & Wang, X.. (2021). Stabilizing Deep Q-Learning with ConvNets and Vision Transformers under Data Augmentation.

---

> > ### Comment · Reviewer_wrjJ · 2022-12-13
> > **Response**
> >
> > I appreciate the author’s careful response, and apologize for the tardiness of this reply.
> >
> > To reply to the author’s comments on weak vs. strong augmentations and generalization, I refer to the author’s own statement in the paper that “Classifying augmentations according to this definition [of weak or strong] is dependent on the task.  For example, cutout color has empirically been shown to be detrimental (‘strong augmentation’) for all tasks in Deep Mind Control Suite (DMC) Tassa et al. (2018), but is a effective (‘weak augmentation’) for Star Pilot in Procgen Cobbe et al. (2019) as shown in Laskin et al.”.  Intuitively, this makes sense, and I agree with the authors - which is what lead me to also conclude that the proposed method would have difficulty being applied out-of-the-box to other domains for which there is no known prior empirical studies or knowledge on the weakness or strength of augmentations.  I believe this still stands, though I also do agree that there are likely some standard/common augmentations that can overall be classified as weak (e.g. shift/crop, as in the cited [3] from the response) - but it seems that the proposed approach would only be executed at its full potential from a task-specific or environment-specific distinction between weak and strong augmentations, which cannot be assumed to be always available.
> >
> > To reply to the author’s comment that “Augmentations…build robustness to visual perturbations on the task trained under” - I agree.  However, what stuck out to me in one of the proposed claims of the paper was the statement that “AugCL…[is]a new method for learning with strong visual augmentations for generalization to unseen environments”.  It appears to me that rather than environments, the authors demonstrate AugCL’s capabilities across tasks in the DMC suite.  I was led to believe by this statement that there would be benefits across different visual domains/environments as well.
> >
> > To reply to the author’s comment that “Splice incorporates distracting backgrounds via image augmentation”, I would say that I fail to see the novelty of this distinction.  Firstly, functionally, the end result is the same as with prior works: that the background has been changed to a distracting scene - it does not seem important _how_ the background was changed, but that it was changed at all.  Furthermore, since Splice was only demonstrated on the same environment (DMC), there does not seem to be a strong argument for the generality of an image augmentation-based background modifier.  This is further supported by the author’s explanation that Splice “mask[s] out all non-relevant parts of the visual observation through a segmentation mask which is available in the simulation” - as Splice relies on access to ground-truth simulation information, why would it be better than previous works that also access the simulation directly?
> >
> > Overall, I believe this is an interesting preliminary investigation, and a well-written paper.  However, I believe there remains more to be done to push this work into a publish-ready submission.  I therefore maintain my original score, and encourage the authors to resubmit after further iterations on the manuscript.

---

### Author Response · Authors · 2022-11-18
**Rebuttal Summary**

We would like to thank the reviewers for the time and effort they put into improving our work. The 2 main concerns of our work were 1) Not enough analysis and experimentation showing the affects of M and 2) not evaluating previous baselines, specifically SVEA and SODA with our novel augmentation Splice. We have addressed both and have included a new table of our results which were added to the paper as well.

$\textbf{Sweep Over M}$:
The mean performance over 3 seeds for each M is given below. Each model is evaluation on 30 episodes.
| Environment | M=0 | M=100k | M=200k | M=300k | M=400k | M=500k |
|-------------|-----|--------|--------|--------|--------|--------|
| Train       | 39  | 877    | 898    | 887    | 852    | 930    |
| Test        | 34  | 892    | 892    | 878    | 842    | 464    |

$\textbf{DMC GB video easy:}$
| Domain, Task       | SODA      | SVEA     | AugCL       | AugCL + SVEA |
|--------------------|-----------|----------|-------------|--------------|
| Walker, Walk       | 625 ± 29  | 882 ±63  | 879 ±35     | **904 ± 29** |
| Walker, Stand      | 875 ± 71  | 969 ±4   | 958 ± 7     | **972 ± 6**  |
| Cartpole, Swingup  | 764 ± 49  | 850 ± 32 | 840 ±27     | **854 ± 9**  |
| Ball In Cup, Catch | 907 ±30   | 963 ± 11 | 959 ± 8     | **967 ± 3**  |
| Finger, Spin       | 888 ± 160 | 975 ± 20 | **983 ± 5** | 975 ± 17     |

$\textbf{DMC GB video hard:}$
| Domain, Task       | SODA      | SVEA     | AugCL        | AugCL + SVEA |
|--------------------|-----------|----------|--------------|--------------|
| Walker, Walk       | 619 ± 25  | 861 ± 59 | 864 ± 34     | **888 ± 30** |
| Walker, Stand      | 872 ± 71  | 960 ± 6  | 959 ± 5      | **962 ± 7**  |
| Cartpole, Swingup  | 594 ± 90  | 776 ± 28 | 742 ± 35     | **784 ± 16** |
| Ball In Cup, Catch | 750 ± 260 | 895 ± 21 | **916 ± 21** | 905 ± 35     |
| Finger, Spin       | 873 ± 163 | 948 ± 20 | 952 ± 24     | **960 ± 17** |

---

### Decision · Program_Chairs · 2023-01-20

**Decision:**

Reject

**Justification For Why Not Higher Score:**

Although the idea of using a curriculum to achieve better CL performance is interesting, the authors did not provide enough clear empirical evidence for it.

**Justification For Why Not Lower Score:**

NA

**Metareview: Summary, Strengths And Weaknesses:**

This paper addresses the problem of generalization to novel visual environments in reinforcement learning. The authors propose a curriculum that incorporates both weak and strong augmentations to improve performance on tasks in the DeepMind Control (DMC) suite. The reviewers agree that the idea of using a curriculum to achieve better results in continuous learning (CL) settings is interesting, they felt like the impact of the curriculum was not established conclusively through the empirical evidence presented in the paper. In particular, the reviewers felt that the effect of the curriculum was confounded by other factors investigated in the paper, which made the overall story less clear. Additionally, the reviewers had concerns about the fact that the results of the proposed AugCL curriculum did not outperform existing methods by a significant margin on the tasks in the DMC suite, and the claim of generalization to unseen environments was not well supported. Although the reviewers appreciated the work the authors put into addressing their concerns during the rebuttal period, they did not feel like the paper was strong enough in its current state to publish at ICLR.